# Absolute Proteome Quantification in the Gas-Fermenting Acetogen *Clostridium autoethanogenum*

Kaspar Valgepea,[a,b] Gert Talbo,[a,c] Nobuaki Takemori,[d] Ayako Takemori,[d] Christina Ludwig,[e] Vishnuvardhan Mahamkali,[f] Alexander P. Mueller,[f] Ryan Tappel,[f] Michael Köpke,[f] Séan Dennis Simpson,[f] Lars Keld Nielsen,[a,c,g] Esteban Marcellin[a,c]

[a]Australian Institute for Bioengineering and Nanotechnology (AIBN), The University of Queensland, St. Lucia, Australia
[b]ERA Chair in Gas Fermentation Technologies, Institute of Technology, University of Tartu, Tartu, Estonia
[c]Queensland Node of Metabolomics Australia, AIBN, The University of Queensland, St. Lucia, Australia
[d]Institute for Promotion of Science and Technology, Ehime University, Ehime, Japan
[e]Bavarian Center for Biomolecular Mass Spectrometry (BayBioMS), Technical University of Munich, Freising, Germany
[f]LanzaTech Inc., Skokie, Illinois, USA
[g]The Novo Nordisk Foundation Center for Biosustainability, Technical University of Denmark, Kongens Lyngby, Denmark

**ABSTRACT** Microbes that can recycle one-carbon ($C_1$) greenhouse gases into fuels and chemicals are vital for the biosustainability of future industries. Acetogens are the most efficient known microbes for fixing carbon oxides $CO_2$ and CO. Understanding proteome allocation is important for metabolic engineering as it dictates metabolic fitness. Here, we use absolute proteomics to quantify intracellular concentrations for >1,000 proteins in the model acetogen *Clostridium autoethanogenum* grown autotrophically on three gas mixtures (CO, $CO+H_2$, or $CO+CO_2+H_2$). We detect the prioritization of proteome allocation for $C_1$ fixation and the significant expression of proteins involved in the production of acetate and ethanol as well as proteins with unclear functions. The data also revealed which isoenzymes are likely relevant *in vivo* for CO oxidation, $H_2$ metabolism, and ethanol production. The integration of proteomic and metabolic flux data demonstrated that enzymes catalyze high fluxes with high concentrations and high *in vivo* catalytic rates. We show that flux adjustments were dominantly accompanied by changing enzyme catalytic rates rather than concentrations.

**IMPORTANCE** Acetogen bacteria are important for maintaining biosustainability as they can recycle gaseous $C_1$ waste feedstocks (e.g., industrial waste gases and syngas from gasified biomass or municipal solid waste) into fuels and chemicals. Notably, the acetogen *Clostridium autoethanogenum* is being used as a cell factory in industrial-scale gas fermentation. Here, we perform reliable absolute proteome quantification for the first time in an acetogen. This is important as our work advances both rational metabolic engineering of acetogen cell factories and accurate *in silico* reconstruction of their phenotypes. Furthermore, this absolute proteomics data set serves as a reference toward a better systems-level understanding of the ancient metabolism of acetogens.

**KEYWORDS** acetogen, gas fermentation, genome-scale metabolic modeling, metabolic modeling, metabolomics, proteomics

Address correspondence to Esteban Marcellin, e.marcellin@uq.edu.au.

The authors declare a conflict of interest. LanzaTech has interest in commercial gas fermentation with *C. autoethanogenum*. Vishnuvardhan Mahamkali, Alexander P. Mueller, Ryan Tappel, Michael Köpke, and Séan Dennis Simpson are employees of LanzaTech.

Increasing concerns about irreversible climate change are accelerating the shift to renewable, carbon-free energy production (e.g., solar, wind, and fuel cells). However, many fuels and chemicals will stay carbon based, and thus, technologies for their production using sustainable and renewable feedstocks are needed to transition toward a circular bioeconomy. Moreover, the rising amount of solid waste produced by human activities (e.g., municipal solid waste and lignocellulosic waste) will further endanger our ecosystems' already critical state. Both challenges can be tackled by using

organisms capable of recycling gaseous one-carbon ($C_1$) waste feedstocks (e.g., industrial waste gases [$CO_2$, CO, and $CH_4$] and syngas from gasified biomass or municipal solid waste [CO, $H_2$, and $CO_2$]) into fuels and chemicals at an industrial scale (1–3).

As we transition into a new bioeconomy, a key feature of global biosustainability will be the capacity to convert carbon oxides into products at an industrial scale. Acetogens are the ideal biocatalysts for this as they use the most energy-efficient pathway, the Wood-Ljungdahl pathway (WLP) (4, 5), for fixing $CO_2$ into the central metabolite acetyl-CoA (6–9) and accept gas (CO, $H_2$, and $CO_2$) as their sole carbon and energy source (5). Indeed, the model acetogen *Clostridium autoethanogenum* is already being used as a cell factory in industrial-scale gas fermentation (3, 10). The WLP is considered the first biochemical pathway on Earth (7, 11–13) and continues to play a critical role in the biogeochemical carbon cycle by fixing an estimated 20% of the global $CO_2$ (6, 14). While biochemical details of the WLP are well described (4, 6, 15), a quantitative understanding of acetogen metabolism is just emerging (16, 17). Notably, recent systems-level analyses of acetogen metabolism have revealed mechanisms behind metabolic shifts (18–21), transcriptional architectures (22, 23), and features of translational regulation (24, 25). However, we still lack an understanding of acetogen proteome allocation through the quantification of proteome-wide intracellular protein concentrations. This fundamental knowledge is required for advancing the rational metabolic engineering of acetogen cell factories and for accurate *in silico* reconstruction of their phenotypes using metabolic models (1, 2).

Quantitative description of an organism's proteome allocation through absolute proteome quantification is valuable in several ways. First, it enables us to understand the prioritization of the energetically costly proteome resources among functional protein categories, metabolic pathways, and single proteins (26, 27). This may also identify relevant proteins with unclear functions and high abundances. Second, some metabolic fluxes can be catalyzed by isoenzymes, and a comparison of their intracellular concentrations can indicate which are likely relevant *in vivo* and are thus targets for genetic perturbation experiments to validate *in vivo* functionalities (28). Third, the integration of absolute proteomics and metabolic flux data enables the estimation of apparent *in vivo* catalytic rates of enzymes ($k_{app}$s) (26, 29), which can be used to identify less-efficient enzymes as targets for improving pathways through metabolic and protein engineering. Absolute proteomics data also contribute to the curation of accurate genome-scale metabolic models.

Absolute proteome quantification is generally performed using label-free mass spectrometry (MS) approaches without spike-in standards (30, 31). The major limitation of this approach is that the accuracy of label-free estimated protein concentrations cannot be determined. Furthermore, the optimal model to convert MS signals (e.g., spectral counts and peak intensities) into protein concentrations remains unknown (32–34). Label-based approaches using stable-isotope-labeled (SIL) spike-ins of endogenous proteins (SIL-proteins) are thus preferred for reliable absolute proteome quantification. This strategy relies on the accurate absolute quantification of a limited set of intracellular proteins (i.e., anchors) using SIL spike-ins to establish a linear correlation between protein concentrations and their measured MS intensities (32). Studies with the latter approach have determined a 1.5- to 2.4-fold error for the label-free estimation of proteome-wide protein concentrations in multiple organisms (28, 35–41).

The aim of our work was to perform reliable absolute proteome quantification for the first time in an acetogen. We employed a label-based MS approach using SIL-protein spike-in standards to quantify SIL-based concentrations for 16 key proteins and label-free-based concentrations for >1,000 *C. autoethanogenum* proteins during autotrophic growth on three gas mixtures. This allowed us to explore global proteome allocation, uncover isoenzyme usage in central metabolism, and quantify regulatory principles associated with estimated $k_{app}$s. Our work provides an important reference data set and advances the systems-level understanding and engineering of the ancient metabolism of acetogens.

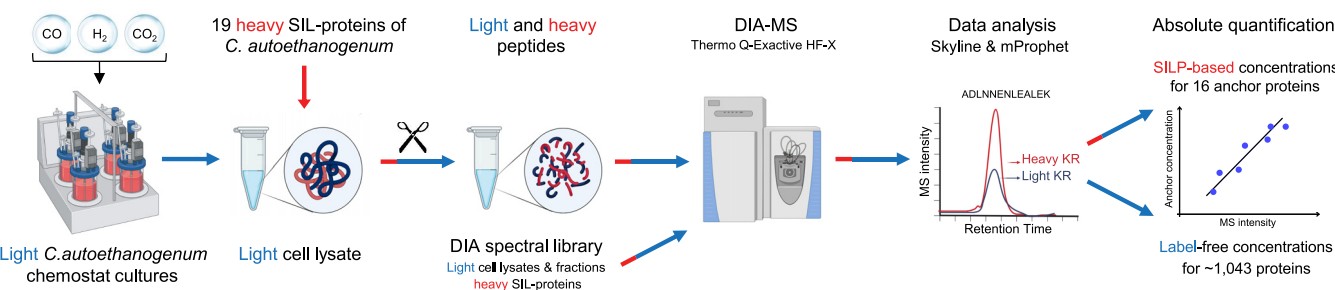

**FIG 1** Absolute proteome quantification framework in *C. autoethanogenum*. Absolute proteome quantification in light (no stable-isotope-labeled [SIL] substrates) autotrophic *C. autoethanogenum* chemostat cultures was built on using 19 synthetic heavy SIL-protein (SILP) spike-in standards and data-independent acquisition (DIA) mass spectrometry (MS) analysis. Culture samples with SIL-protein spike-ins and samples for the DIA spectral library were analyzed by DIA MS. Subsequent stringent data analysis allowed the quantification of intracellular concentrations for 16 key *C. autoethanogenum* proteins using light-to-heavy ratios between endogenous and spike-in DIA MS intensities. These 16 key proteins were further used as anchor proteins for the label-free estimation of ~1,043 protein concentrations through establishing a linear correlation between protein concentrations and their measured MS intensities. KR, lysine-arginine. Some parts were created with BioRender.com. See also Table S1 in the supplemental material.

## RESULTS

**Absolute proteome quantification framework in the model acetogen *C. autoethanogenum*.** We performed absolute proteome quantification from autotrophic steady-state chemostat cultures of *C. autoethanogenum* grown on three different gas mixtures: CO, syngas ($CO+CO_2+H_2$), or $CO+H_2$ (termed "high-$H_2$ CO"), described previously (18, 19). Briefly, four biological cultures of each gas mixture were grown anaerobically on a chemically defined medium at 37°C, pH 5, and a dilution rate of ~1 day$^{-1}$ (specific growth rate of ~0.04 h$^{-1}$), without the use of heavy SIL substrates. Among the previously quantified differences in culture phenotypes (18, 19), most striking were the significantly increased and decreased carbon fluxes to ethanol and $CO_2$, respectively, with the increasing supply of $H_2$. The absolute proteome quantification framework (Fig. 1) was built on using 19 synthetic heavy SIL variants of key *C. autoethanogenum* proteins covering central metabolism (see Table S1 in the supplemental material). The SIL-protein standards were spiked in for the quantification of the intracellular concentrations of their endogenous light counterparts. This framework ensures accurate absolute quantification compared to commonly used peptide spike-ins. Spiking cell lysates with protein standards before sample cleanup and protein digestion accounts for errors accompanying these critical steps (30, 31, 42, 43). Furthermore, the selection of peptides ensuring accurate absolute protein quantification without prior MS data is challenging as it is difficult to predict which peptides "fly" well (30, 31, 42, 43). In contrast, all proteotypic peptides from a protein spike-in can be used for quantification.

We synthesized heavy-labeled lysine and arginine SIL-proteins using a cell-free wheat germ extract platform as described previously (18, 44, 45) and quantified standard stocks using parallel-reaction monitoring (PRM) MS. Next, proteins were extracted from culture samples using an optimized protocol maximizing the extraction yield (18) followed by spike-in of the 19 heavy SIL-proteins into light cell lysates. We then used a data-independent acquisition (DIA) MS approach (46) to quantitate 1,243 proteins of *C. autoethanogenum* across 12 samples (quadruplicate cultures of three gas mixtures) using a comprehensive spectral library consisting of whole-cell lysates, lysate fractions, and spike-in SIL-proteins. Finally, we quantified intracellular concentrations for 16 key *C. autoethanogenum* proteins using light-to-heavy (L/H) ratios between endogenous and spike-in DIA MS intensities and further used these 16 as anchor proteins for the label-free estimation of ~1,043 protein concentrations by establishing a linear correlation between protein concentrations and their measured MS intensities (32). We express intracellular protein concentrations in nanomoles of protein per gram of dry cell weight (nmol/gDCW).

**Absolute quantification of 16 anchor protein concentrations.** To ensure high-confidence absolute quantification of anchor protein concentrations from the DIA MS

**TABLE 1** Anchor protein concentrations in *C. autoethanogenum* chemostat cultures

| Protein ID[c] | Name[b] | Description of protein product[c] | No. of peptides[d] | Protein concentration (nmol/gDCW)[a] | | |
|---|---|---|---|---|---|---|
| | | | | CO | Syngas | High-H$_2$ CO |
| CAETHG_RS00440 | AOR1 | Aldehyde ferredoxin oxidoreductase | 16 | 270 ± 15 | 211 ± 19 | 290 ± 61 |
| CAETHG_RS00580 | IlvC | Ketol-acid reductoisomerase | 7 | 186 ± 8 | 196 ± 27 | 190 ± 42 |
| CAETHG_RS00590 | IlvB3 | Acetolactate synthase, large subunit | 2 | 13 ± 1 | 7 ± 1 | 104 ± 26 |
| CAETHG_RS02630 | Adh3 | NADPH-dependent butanol dehydrogenase | 2 | 18 ± 4 | 8 ± 2 | 2 ± 0.1 |
| CAETHG_RS07815 | AcsD | Corrinoid iron-sulfur protein part 2 | 7 | 633 ± 44 | 509 ± 41 | 703 ± 40 |
| CAETHG_RS07830 | MetF | Methylene-THF reductase | 5 | 254 ± 16 | 233 ± 16 | 280 ± 36 |
| CAETHG_RS07840 | FolD | Methylene-THF dehydrogenase | 4 | 114 ± 4 | 103 ± 10 | NQ |
| CAETHG_RS07845 | FchA | Methenyl-THF cyclohydrolase | 2 | 308 ± 30 | 221 ± 7 | 313 ± 68 |
| CAETHG_RS07850 | Fhs | Formate-THF ligase | 13 | 953 ± 109 | 707 ± 49 | 1,122 ± 107 |
| CAETHG_RS07861 | AcsA (CODH) | Carbon monoxide dehydrogenase (CODH) | 1 | 122 ± 32 | 119 ± 21 | 145 ± 37 |
| CAETHG_RS09495 | RplA | 50S ribosomal protein L1 | 2 | 88 ± 1 | 103 ± 12 | 79 ± 12 |
| CAETHG_RS13725 | FdhA | Formate dehydrogenase subunit alpha | 9 | 62 ± 4 | 41 ± 5 | NQ |
| CAETHG_RS13765 | HytA | Hydrogenase, Fe only | 8 | 72 ± 1 | 55 ± 6 | 94 ± 20 |
| CAETHG_RS14890 | PFOR | Pyruvate:ferredoxin (flavodoxin) oxidoreductase | 19 | 133 ± 7 | 178 ± 13 | 45 ± 9 |
| CAETHG_RS16490 | Pta | Phosphate acetyltransferase | 6 | 81 ± 4 | 89 ± 4 | 81 ± 4 |
| CAETHG_RS16495 | AckA | Acetate kinase | 3 | 58 ± 1 | 71 ± 8 | 51 ± 6 |

[a]Protein concentration data represent averages ± standard deviations from four biological replicate cultures. gDCW, gram of dry cell weight; THF, tetrahydrofolate; NQ, not quantified (see Materials and Methods).
[b]Generated during KEGG Orthology (KO) analysis using BlastKOALA (see Materials and Methods) or manually edited.
[c]From the NCBI annotation of the sequence under GenBank accession number NC_022592.1 (see Materials and Methods) or manually edited.
[d]Number of peptides used for quantification (106 in total).

data, we employed stringent criteria on top of the automated mProphet peak-picking algorithm (47) within the software Skyline (48). We also performed a dilution series experiment for each SIL-protein to increase accuracy (see Materials and Methods for details). Briefly, we kept only peaks with Gaussian shapes and without interference and precursors with the highest Skyline quality metrics. Importantly, only peptides whose signals were above the lower limit of quantification (LLOQ) and within the linear dynamic quantification range in the dilution series experiment were used for anchor protein quantification (Table S2). We thus used 106 high-confidence peptides for the absolute quantification of 16 anchor protein concentrations (Table 1; see also Fig. 5). The high confidence of the intracellular concentrations for these key *C. autoethanogenum* proteins of central metabolism is supported both by the low average 11% coefficient of variation (CV) between biological quadruplicate cultures (Table 1) and the average 22% CV between different peptides of single proteins (Table S2).

**Label-free estimation of proteome-wide protein concentrations.** Both high-quality proteomics data and suitable anchor proteins are required for reliable label-free absolute proteome quantification. Our proteome-wide DIA MS data were highly reproducible, with an average Pearson correlation coefficient (*R*) of 0.99 between biological replicates (Fig. 2A and Fig. S1). We also found our anchor proteins suitable as their concentrations spanned across 3 orders of magnitude, and the summed mass accounted for ~1/3 of the peptide mass injected into the mass spectrometer (Table 1 and Fig. 2B). We used the 16 anchor proteins (with 106 peptides) to determine the optimal label-free quantification model with the best linear fit between anchor protein concentrations and their measured DIA MS intensities using the aLFQ R package (49) as described previously for SWATH MS (28) (Fig. 2C). Notably, we detected an average 1.5-fold cross-validated mean fold error (CV-MFE) (bootstrapping) for the label-free estimated anchor protein concentrations across samples (Fig. 2D). The errors were distributed normally (Fig. S2), with an average 95% confidence interval (CI) of 0.3 (Fig. 2D). We then applied the optimal label-free quantification model to estimate ~1,043 protein concentrations in *C. autoethanogenum* (Table S3).

Prior to the detailed analysis of proteome-wide protein concentrations, we further evaluated our label-free data accuracy beyond the 1.5-fold CV-MFE determined as described above. First, the total proteome mass (1.2 ± 0.1 μg [average ± standard

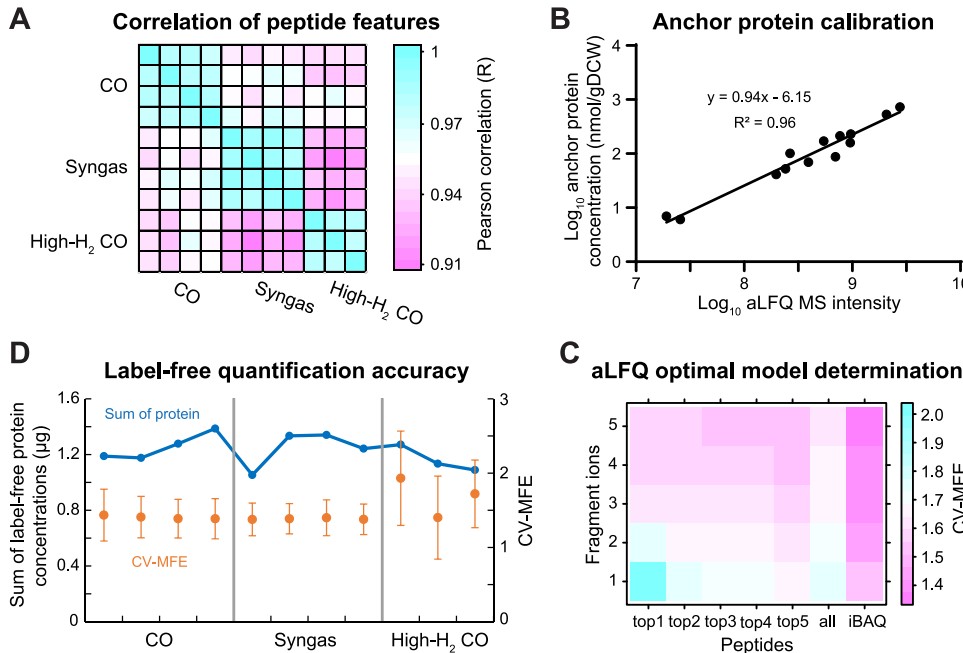

**FIG 2** Label-free estimation of proteome-wide protein concentrations. (A) Correlation of peptide mass spectrometry (MS) feature intensities between biological replicate cultures of the three gas mixtures. (B) Linear correlation between anchor protein concentrations and their measured MS intensities for one syngas culture. gDCW, gram of dry cell weight; aLFQ, absolute label-free quantification. (C) Errors of different label-free quantification models for the linear fit between anchor protein concentrations and their measured MS intensities determined by bootstrapping using the aLFQ R package (49) for one syngas culture. CV-MFE, cross-validated mean fold error. (D) Label-free quantification error of the optimal model (orange) and total proteome mass (blue) across samples. Error bars denote 95% CIs. See also Fig. S1 and S2 and Table S2 in the supplemental material.

deviation]) closely matched the 1-$\mu$g peptide mass injected into the mass spectrometer (Fig. 2D), translating into 0.54 ± 0.05 g of protein/gDCW. The data were also supported by a strong correlation between estimated protein concentrations and expected stoichiometries for equimolar and nonequimolar protein complexes (Fig. 3). Note that given that we did not use a specific method for membrane proteins, we did not include membrane protein complexes, including the Rnf complex. Notably, the absolute protein concentrations of syngas cultures correlated well ($R = 0.65$) with their respective absolute transcript expression levels determined previously (19) (Fig. S3). This result is similar to the correlations of absolute data seen in other steady-state cultures (26, 50). Altogether, we present the first absolute quantitative proteome data set for a gas-fermenting acetogen that includes SIL-based concentrations for 16 key proteins and label-free estimates for over 1,000 *C. autoethanogenum* proteins during growth on three gas mixtures.

**C$_1$ fixation dominates global proteome allocation.** Global proteome allocation among functional gene classifications was explored using proteomaps (27) and KEGG Orthology identifiers (KO IDs) (51). The "treemap" structure defining the four-level hierarchy of our proteomaps (Table S4) also included manually curated categories to accurately reflect acetogen metabolism (e.g., C$_1$ fixation/WLP and hydrogenases). As expected for the autotrophic growth of an acetogen, the C$_1$ fixation (Fig. 4) or WLP (Fig. S4) categories dominated the proteome allocation across the three gas mixtures with an ~1/3 fraction, compared to carbohydrate metabolism or glycolysis/gluconeogenesis. Notably, the data show that two genes, dihydrolipoamide dehydrogenase (LpdA; CAETHG_RS07825) and glycine cleavage system H protein (GcvH; RS07795), encoded by the WLP gene cluster were translated at very high levels (Fig. 4). This is important as both have unknown functions in *C. autoethanogenum* metabolism. The significant investment in the expression of proteins involved in acetate and ethanol

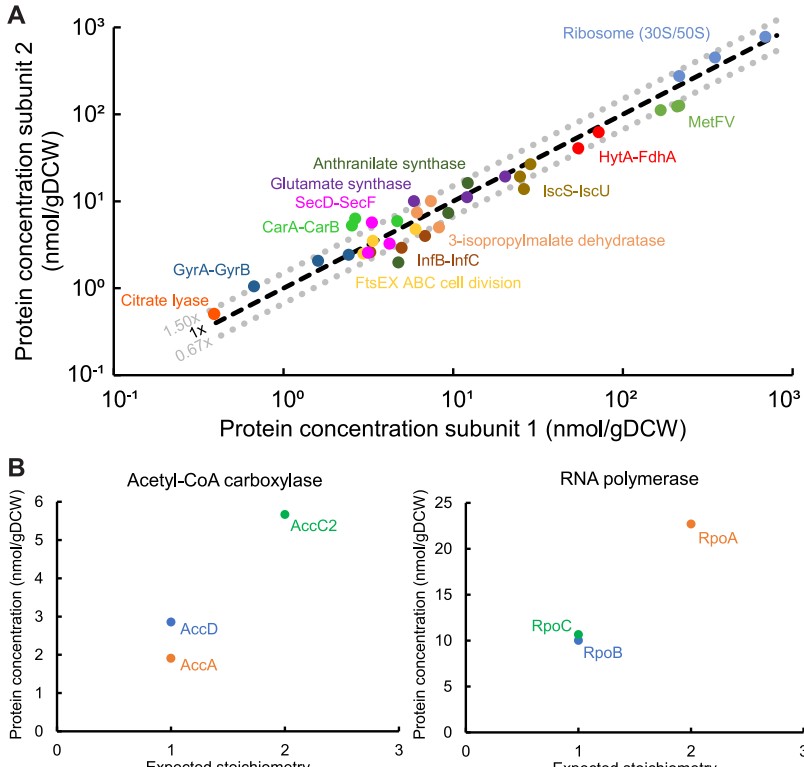

**FIG 3** Strong correlation between protein concentrations and expected stoichiometries for protein complexes. (A) Equimolar protein complexes. Gray dotted lines denote the average 1.5-fold cross-validated mean fold error (CV-MFE) of label-free protein concentrations. Label-free protein concentrations are plotted, except for the HytA-FdhA complex, which was quantified using stable-isotope-labeled protein spike-ins. Data points of the same color represent gas mixtures. Ribosome (30S/50S) stands for 16 and 26 proteins belonging to the 30S and 50S ribosome subunits, respectively. (B) Nonequimolar protein complexes. Data are averages from CO cultures. See Materials and Methods for details on expected protein complex stoichiometries. See Table S3 in the supplemental material for gene/protein identifiers, proposed names, descriptions, and label-free data. See Table 1 for HytA-FdhA data. gDCW, gram of dry cell weight. See also Fig. S4.

production (Fig. S4) is consistent with the 1/3 to 2/3 of fixed carbon channeled into these two growth by-products across the three gas mixtures (18, 19). The 11% proteome fraction of the category translation (Fig. 4) is expected for cells growing at a specific growth rate of ~0.04 h$^{-1}$ based on absolute proteomics data from *Escherichia coli* (26, 39, 52). The notable proteome allocations for amino acid metabolism and particularly the high abundance of ketol-acid reductoisomerase (IlvC; RS00580) are surprising since the metabolic fluxes through 2,3-butanediol and branched-chain amino acid pathways were low under these growth conditions (18, 19). In addition, numerous proteins with unknown or unclear functions (in gray in the proteomaps) are highly expressed (e.g., RS12590, RS08610, and RS08145), highlighting the need for global mapping of genotype-phenotype relationships in acetogens. In general, the proteome allocations were highly similar among the three gas mixtures (Table S3). This result is unsurprising given the few relative protein expression differences detected previously among the same cultures (18).

**Enzyme usage revealed in central metabolism.** Next, we focused on uncovering enzyme usage in acetogen central metabolism (Fig. 5). This contains enzymes of the WLP; acetate, ethanol, and 2,3-butanediol production pathways; hydrogenases; and the Nfn transhydrogenase, which together carry >90% of the carbon and most of the redox flow in *C. autoethanogenum* (18–20). Multiple metabolic fluxes in these pathways can be catalyzed by isoenzymes, and absolute proteomics data can indicate which of the isoenzymes are likely relevant *in vivo*. While the carbon monoxide dehydrogenase

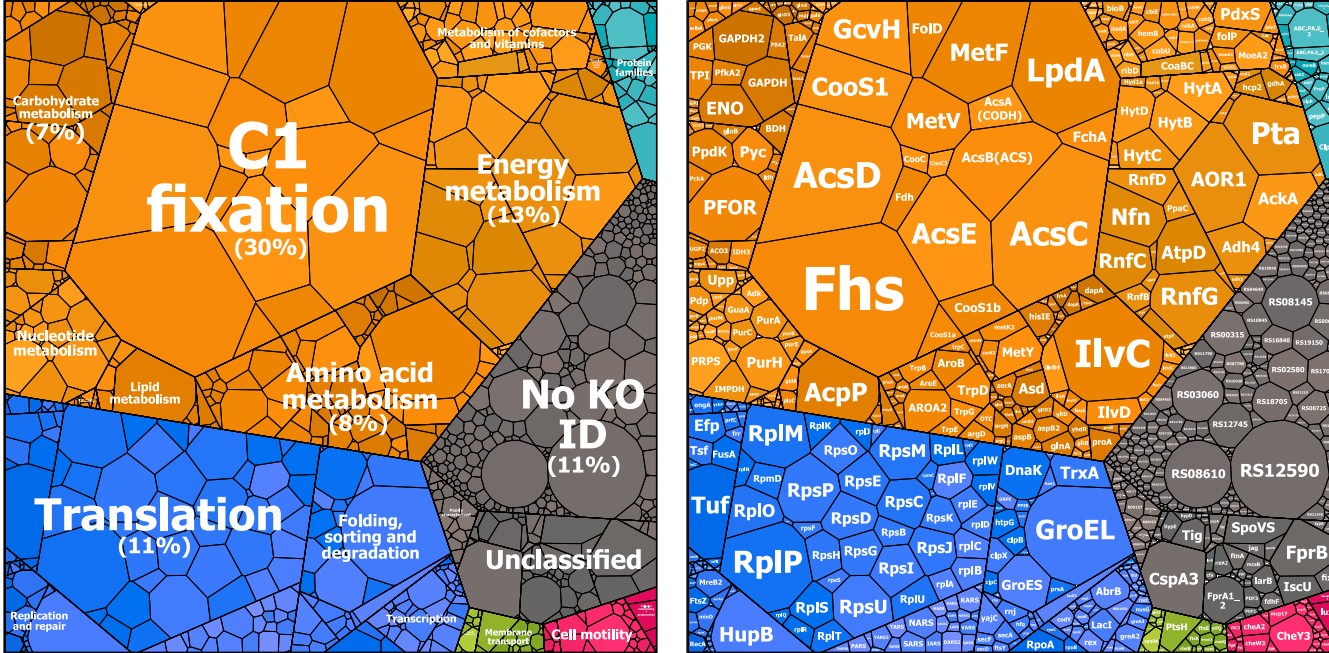

**FIG 4** Proteomaps uncover global proteome allocation. The left proteomap shows proteome allocations among functional gene classification categories (KEGG Orthology identifiers [KO IDs] [51]) at level 2 of the four-level "treemap" hierarchy structure (see Table S4 in the supplemental material). The right proteomap shows proteome allocations at the level of single proteins (level 4 of treemap). See Fig. S4 for proteomaps of levels 1 and 3 of treemap. The area of the tile is proportional to the protein concentration. Colors denote level 1 categories of treemap. Proteomaps visualize average concentrations of syngas cultures, while category percentages are averages from three gas mixtures (shown for categories with a fraction of >5%). See Table S3 for gene/protein identifiers, proposed names, descriptions, and label-free protein concentrations. See also Fig. S4 and Tables S3 and S4.

(CODH) AcsA (RS07861-RS07862) that forms the bifunctional CODH/ACS complex with the acetyl-CoA synthase (53) (AcsB; RS07800) is essential for *C. autoethanogenum* growth on gas, as confirmed in mutagenesis studies (54), the higher concentrations of the dispensable monofunctional CODH CooS1 (RS14775) suggest that it may also play a role in CO oxidation (Fig. 4 and 5), in addition to $CO_2$ reduction (54). Additionally, our proteomics data show a high abundance of the primary acetaldehyde:ferredoxin oxidoreductase (AOR1; RS00440), and this supports the emerging understanding that in *C. autoethanogenum*, ethanol is dominantly produced using AOR1 activity via acetate instead of directly from acetyl-CoA via acetaldehyde using mono- or bifunctional activities (18, 19, 55, 56) (Fig. 5). Furthermore, the data suggest that the specific alcohol dehydrogenase (Adh4; RS08920) is responsible for reducing acetaldehyde to ethanol, a key reaction in terms of carbon and redox metabolism. The high abundance of the electron-bifurcating hydrogenase HytA-E (HytA through -E) complex (RS13745–70) compared to alternative hydrogenases confirms that it is the main $H_2$ oxidizer (57, 58) (Fig. 5). This is consistent with the fact that in the presence of $H_2$, all the $CO_2$ fixed by the WLP is reduced to formate using $H_2$ by the HytA-E and formate dehydrogenase (FdhA; RS13725) enzyme complex activity (18, 19). Despite the proteomics evidence, genetic perturbations are required to determine the condition-specific *in vivo* functionalities of isoenzymes in acetogens unequivocally.

The overall most abundant protein was the formate-tetrahydrofolate ligase (Fhs; RS07850), a key enzyme in the WLP (Fig. 4 and 5). Despite the high abundance, its expression might still be rate limiting (see below). Another key enzyme for acetogens is AcsB because of its essentiality for acetyl-CoA synthesis by the CODH/ACS complex. AcsB is linked to the WLP by the corrinoid iron-sulfur proteins AcsC (RS07810) and AcsD (RS07815) that supply the methyl group to AcsB. Interestingly, the ratio of AcsCD to AcsB increased from 1.7 (CO) to 2.3 (syngas) to 2.9 (high-$H_2$ CO), suggesting that the primary role of the CODH/ACS complex shifted from CO oxidation toward acetyl-CoA synthesis. This could be caused by the increased $H_2$ uptake replacing the supply of

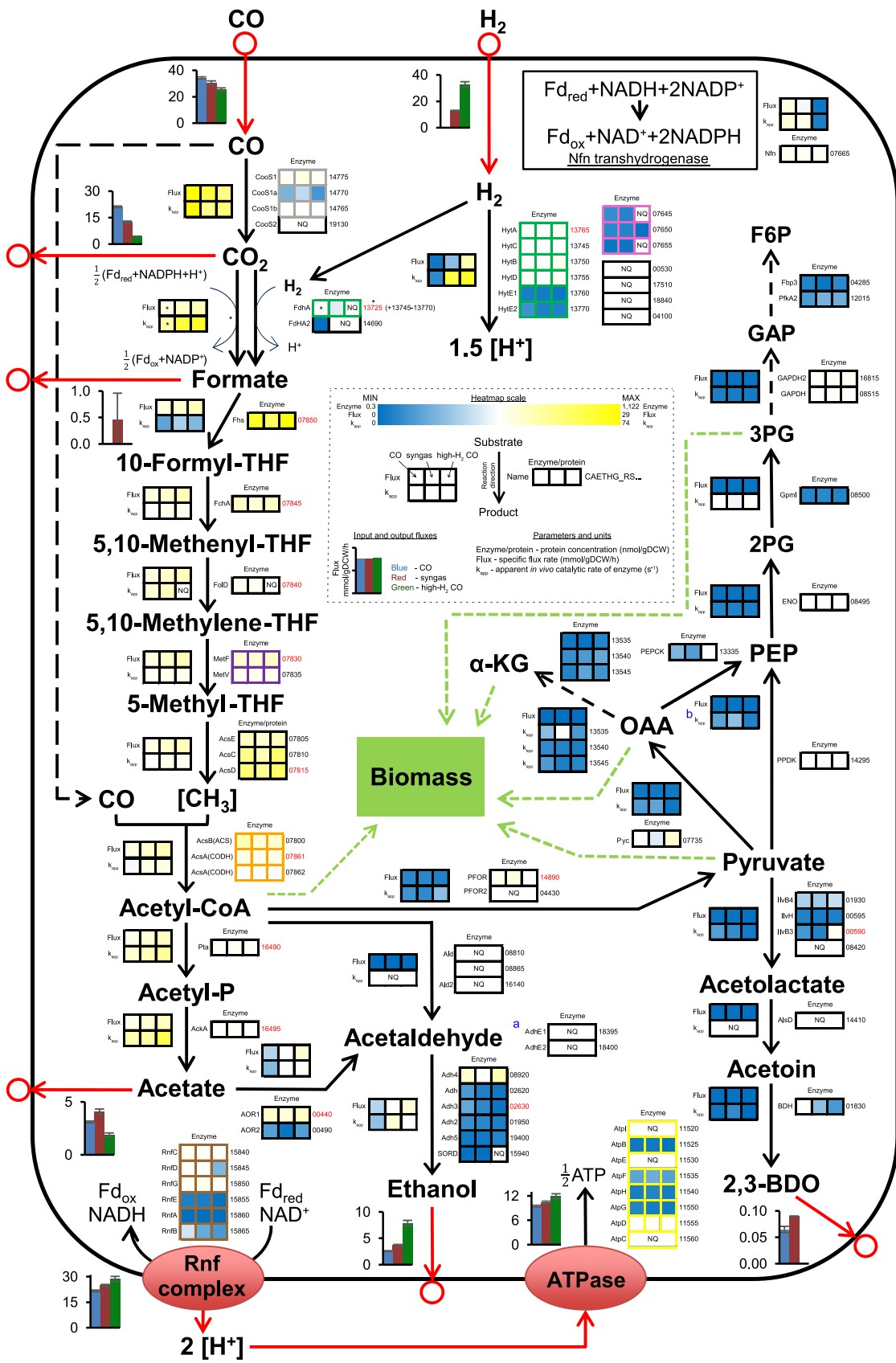

**FIG 5** Quantitative systems-level view of acetogen central metabolism. Enzyme concentrations (nmol/gDCW [gram of dry cell weight]), apparent *in vivo* catalytic rates of enzymes ($k_{app}$) (s$^{-1}$), and metabolic flux rates (mmol/gDCW/h) are shown for *C. autoethanogenum*

reduced ferredoxin from CO oxidation or by the decreasing specific CO uptake rate (Fig. 5). Concurrently, the Nfn transhydrogenase (RS07665) levels that act as a redox valve in acetogens (20) are maintained high (Fig. 5), potentially to rapidly respond to redox perturbations. We conclude that absolute quantitative proteomics can significantly contribute to a systems-level understanding of metabolism, particularly in less-studied organisms.

**Integration of absolute proteomics and flux data yields *in vivo* enzyme catalytic rates.** Absolute proteomics data enable the estimation of intracellular catalytic working rates of enzymes when metabolic flux rates are known (26, 29). We thus calculated the apparent *in vivo* catalytic rates of enzymes, denoted $k_{app}$ ($s^{-1}$) (26), as the ratio of the specific metabolic flux rate (mmol/gDCW/h) determined previously (18) to the protein concentration (nmol/gDCW) determined here for the same *C. autoethanogenum* CO, syngas, and high-$H_2$ CO cultures (see Materials and Methods). For reactions assigned with isoenzymes, the enzyme with the highest abundance ranking was assumed to solely catalyze the flux. This produced $k_{app}$ values for 48 enzymes/complexes using label-free protein concentrations and for 13 of those also using anchor protein concentrations (Fig. 5 and 6 and Table S5). The first two critical steps for carbon fixation in the methyl branch of the WLP (i.e., CO to formate) are catalyzed at high rates (Fig. 5). Notably, FdhA showed a $k_{app}$ of $\sim$30 $s^{-1}$ for $CO_2$ reduction without $H_2$ during growth on CO only, which is similar to *in vitro* $k_{cat}$ data for formate dehydrogenases ($\sim$28 to 47 $s^{-1}$) in other chemolithoautotrophs (59, 60). Interestingly, the next step of formate reduction was catalyzed potentially by a less-efficient enzyme, Fhs, as its $k_{app}$ of $\sim$3 $s^{-1}$ is significantly lower than those of other WLP enzymes (Fig. 5). At the same time, Fhs is the most abundant protein in the *C. autoethanogenum* proteome (Fig. 4). Overall, enzymes catalyzing reactions in high-flux pathways such as the WLP and acetate and ethanol production have higher $k_{app}$s than those downstream from the conversion of acetyl-CoA to pyruvate (Fig. 5). Indeed, enzymes catalyzing high metabolic fluxes in *C. autoethanogenum* have both higher concentrations and higher catalytic rates than enzymes catalyzing lower fluxes as both specific flux rates and enzyme concentrations (Kendall's $\tau$ = 0.56; $P = 5 \times 10^{-9}$) and flux and $k_{app}$ ($\tau$ = 0.45; $P = 2 \times 10^{-6}$) were significantly correlated (Fig. 6A), as seen previously for other organisms (26, 61).

Having acquired absolute proteomics data for *C. autoethanogenum* growth on three gas mixtures with different metabolic flux profiles also allowed us to determine the impact of the change in the enzyme concentration and its catalytic rate for adjusting metabolic flux rates. Two extreme examples are the reactions catalyzed by the HytA-E (Fig. 6B) and Nfn (Fig. 5) complexes where flux adjustments were accompanied by large changes in $k_{app}$s rather than in enzyme concentrations. Flux changes in high-flux pathways such as the WLP and acetate and ethanol production also coincided mainly with $k_{app}$ changes (Fig. 6B). This principle seems to be dominant in *C. autoethanogenum* as 90% of the flux changes were not regulated through enzyme concentrations (i.e., posttranslational regulation) (Table S6) when comparing all statistically significant flux changes among the three gas mixtures with the respective enzyme expression changes (see Materials and Methods). We acknowledge that this analysis is sensitive to the accura-

**FIG 5** Legend (Continued)

steady-state chemostat cultures grown on three gas mixtures. See the dashed inset for bar chart and heatmap details. Enzyme concentration and $k_{app}$ data are averages from biological replicates. Proteins forming a complex are highlighted with nonblack borders (FdhA forms a complex with HytA-E for direct $CO_2$ reduction with $H_2$; CooS1 is expected to form a complex with CooS1a and -b as they are encoded from the same operon). For reactions with isoenzymes, $k_{app}$ is for the enzyme with the highest concentration ranking (top location on the enzyme heatmap) (see Materials and Methods for details). Flux data reported previously by Valgepea et al. (18) are averages from biological replicates, and error bars denote standard deviations. Arrows show the direction of calculated fluxes; red arrows denote uptake or secretion. Gene/protein identifiers to the right of the enzyme concentration heatmaps are preceded by CAETHG_RS, and red font denotes the concentrations determined using stable-isotope-labeled (SIL) protein spike-in standards (i.e., anchor proteins). An asterisk denotes data for redox-consuming $CO_2$ reduction to formate solely by FdhA without the use of $H_2$ during growth on CO. [a]Bifunctional acetaldehyde/alcohol dehydrogenase (acetyl-CoA→ethanol). [b]Flux into phosphoenolpyruvate (PEP) from oxaloacetate (OAA) and pyruvate is merged, and the $k_{app}$ is for PEPCK. See Table S3 in the supplemental material for gene/protein identifiers, proposed names, descriptions, and label-free protein concentrations. See Table 1 for anchor protein concentrations. See Table S5 for $k_{app}$ and flux data. See reference 18 for cofactors of reactions and metabolite abbreviations. NQ, not quantified. See also Tables S3 and S5.

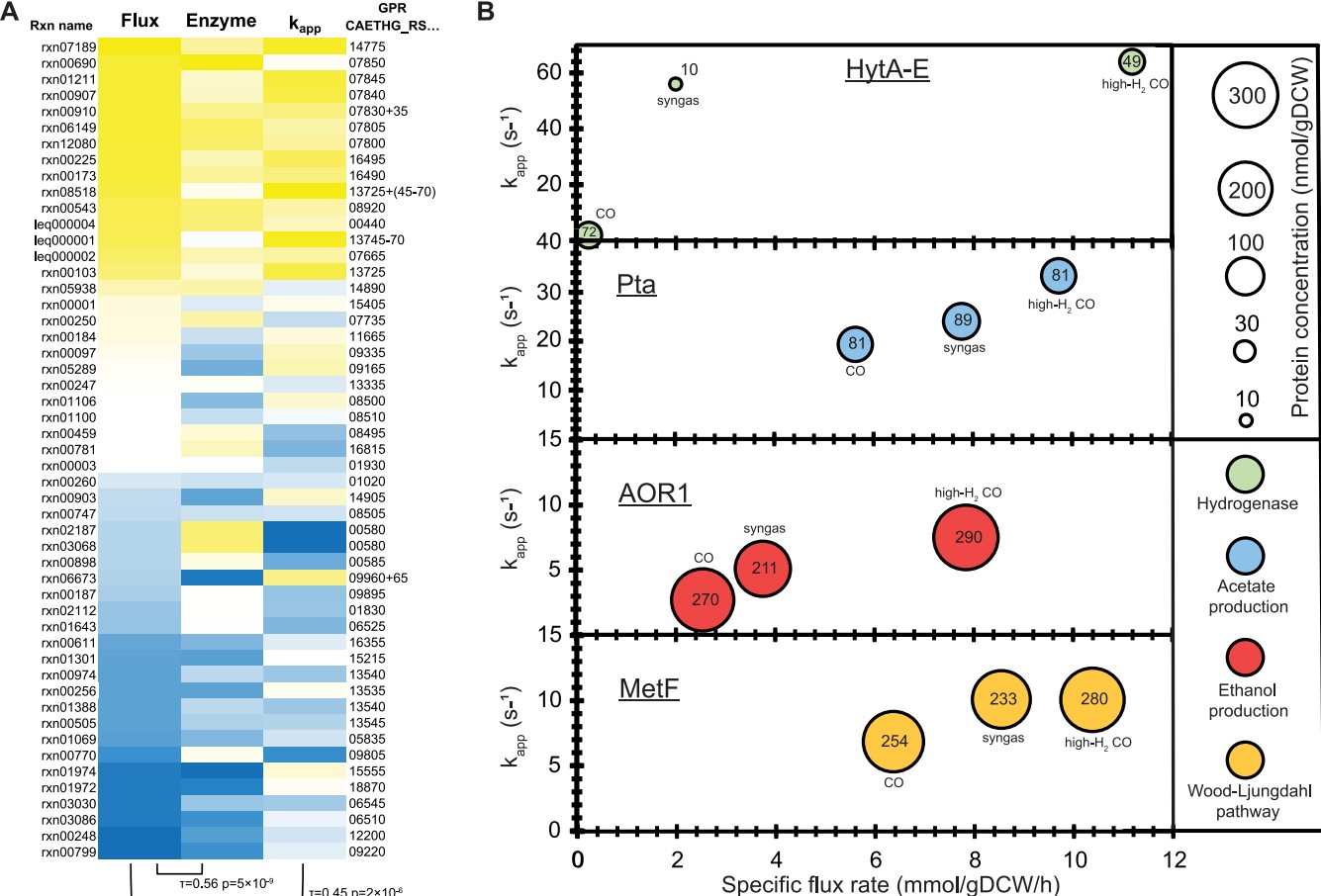

**FIG 6** Regulatory principles of apparent *in vivo* catalytic rates of enzymes ($k_{app}$s) and metabolic flux throughput. (A) Enzymes catalyzing higher metabolic flux rates have both higher concentrations and higher $k_{app}$s. Yellow and blue denote high and low values, respectively. Kendall's $\tau$ correlations with significant $P$ values between the respective pairs are shown below the heatmap. See Table S5 in the supplemental material for flux rate, enzyme concentration, and $k_{app}$ data and for descriptions of reaction names (Rxn names) and gene-protein-reaction (GPR) associations. (B) Control of metabolic flux throughput through $k_{app}$ changes for high-flux pathways. See also Fig. 5. gDCW, gram of dry cell weight. See also Table S6.

cies of both flux simulations and proteomic measurements and also to potential inconsistencies in gene-protein-reaction (GPR) associations for a nonconventional organism.

## DISCUSSION

Acetogen gas fermentation allows the recycling of gaseous and solid waste feedstocks into fuels and chemicals (1, 2). Although the systems-level understanding of acetogen metabolism has recently improved (16, 17), a quantitative description of acetogen proteome allocation was missing. Thus, we performed absolute proteome quantification in autotrophic *C. autoethanogenum* cultures. Our results propose enzyme targets for improving pathways through protein engineering and call for the mapping of genotype-phenotype relationships in acetogens to advance their metabolic engineering into superior cell factories.

Our absolute proteome quantification framework relied on SIL-protein spike-in standards and DIA MS analysis to ensure high confidence in the determined intracellular concentrations for 16 key *C. autoethanogenum* proteins. We further used these proteins as anchor proteins for the label-free estimation of >1,000 protein concentrations. This enabled us to determine the optimal label-free quantification model for our data to infer protein concentrations from MS intensities, which remains unknown in common label-free approaches not utilizing spike-in standards (32, 33). More importantly, label-free estimated protein concentrations using the latter approach are questionable as their accuracy cannot be determined. We determined an excellent average error of 1.5-fold for our label-free estimated protein concentrations based on 16 anchor

proteins and a bootstrapping approach. This error is in the same range as those described in previous studies using SIL spike-in standards for absolute proteome quantification (28, 35–41). Furthermore, we also observed a good match both between estimated and injected proteome masses into the mass spectrometer and between protein concentrations and expected protein complex stoichiometries. We conclude that label-free estimation of proteome-wide protein concentrations using SIL-protein spikeins and state-of-the-art MS analysis is reasonably accurate.

Quantification of acetogen proteome allocation during autotrophic growth expectedly showed the prioritization of proteome resources for fixing carbon through the WLP, in line with transcript expression data in *C. autoethanogenum* (19, 56). The allocation of one-third of the total proteome for $C_1$ fixation is higher than the proteome allocation for carbon fixation through glycolysis during the heterotrophic growth of other microorganisms (26, 36). High abundances of other key enzymes of acetogen central metabolism were also expected as the WLP, the acetate and ethanol production pathways, hydrogenases, and the Nfn transhydrogenase carry >90% of the carbon and most of the redox flow in *C. autoethanogenum* (18–20). However, the very high expression levels of two genes, LpdA and GcvH, of the WLP gene cluster with unknown functions in *C. autoethanogenum* are striking. The functional cooperation of the glycine synthase-reductase pathway that involves LpdA and GcvH and the WLP for the autotrophic growth of *Clostridium drakei* was recently demonstrated (62), raising the question of whether the genes might play a similar role in *C. autoethanogenum*. Since many other proteins with unknown or unclear functions were also highly abundant, global mapping of genotype-phenotype relationships in acetogens is much needed.

The *in vivo* functionalities of isoenzymes are not clear for multiple key metabolic fluxes in acetogen central metabolism, and absolute proteomics data can indicate which isoenzymes are likely relevant. The oxidation of CO or the reduction of $CO_2$ is a fundamental step for all acetogens and is known to be catalyzed by three CODHs in *C. autoethanogenum* (54). Although only AcsA that forms the bifunctional CODH/ACS complex with the acetyl-CoA synthase (53) is essential for growth on gas (54), we detected higher concentrations of the monofunctional CODH CooS1. Intriguingly, the deletion of *cooS1* in *C. autoethanogenum* leads to higher ethanol-to-acetate ratios during growth on CO and a doubling of maximal biomass levels without an apparent lag phase on $CO_2+H_2$ (54). Concurrently, our data suggest that the prioritization of CODH/ACS activity between CO oxidation and acetyl-CoA synthesis is sensitive to $H_2$ availability. Thus, further studies are required to decipher the condition-dependent functionalities of CODHs. In addition to CODHs, the biochemical understanding of ethanol production is important in terms of both carbon and redox metabolism. Our data confirm that in *C. autoethanogenum*, ethanol is predominantly produced via acetate by AOR1 (18, 19, 55, 56) and, more importantly, indicate for the first time that AOR1 activity is followed by Adh4 (previously characterized as butanol dehydrogenase [63]) for the reduction of acetaldehyde to ethanol. These observations call for large-scale genetic perturbation experiments to determine unequivocally the condition-specific *in vivo* functionalities of isoenzymes in acetogens.

Absolute proteomics data offer a unique opportunity to estimate apparent *in vivo* catalytic rates of enzymes ($k_{app}$) (26, 29) if metabolic flux data are also available. These data are particularly valuable for more-accurate *in silico* reconstructions of phenotypes using protein-constrained genome-scale metabolic models (64, 65). While *in vitro* $k_{cat}$ and *in vivo* $k_{app}$ data generally correlate (29), models using maximal $k_{app}$ values show better predictions of protein abundances (66). Furthermore, information on $k_{app}$s can infer less-efficient enzymes as targets for improving pathways through metabolic and protein engineering. For example, protein engineering of Fhs (catalyzing formate reduction) might improve WLP throughput and carbon fixation since its $k_{app}$ was significantly lower than those of other pathway enzymes. At the same time, the large changes in $k_{app}$s of the abundant electron-bifurcating hydrogenase HytA-E and the Nfn transhydrogenase complexes indicate the capacity for the cells to rapidly respond to

$H_2$ availability and redox perturbations, which may be critical for the metabolic robustness of acetogens (20). Overall, we detected both higher concentrations and higher $k_{app}$s for enzymes catalyzing higher metabolic fluxes, which is believed to arise from an evolutionary push toward reducing protein production costs for enzymes carrying high flux (61). It remains to be seen whether $k_{app}$s for enzymes catalyzing both glycolysis and gluconeogenesis fluxes would increase with the heterotrophic growth of *C. autoethanogenum* and how they would compare with the respective $k_{app}$s for an organism that has optimized catabolism of sugars, like *E. coli* (26). The observation that 90% of flux changes in *C. autoethanogenum* were not regulated through changes in enzyme concentrations is not surprising for a metabolism that operates at the thermodynamic edge of feasibility (16, 17) since the posttranslational regulation of fluxes is energetically the least costly. That being said, we note the sensitivity of our analysis to accuracies of experimental measurements and also to GPR associations despite their manual curation. Further research is needed to identify which mechanism from posttranslational protein modification, allosteric regulation, or substrate concentration change is responsible for the posttranslational regulation of fluxes. Interestingly, recent work demonstrates the relevance of both protein acetylation and intracellular metabolite levels for the regulation of acetogen metabolism (20, 67, 68).

We have produced the first absolute proteome quantification in an acetogen and thus provided an understanding of global proteome allocation, isoenzyme usage in central metabolism, and regulatory principles of *in vivo* enzyme catalytic rates. This fundamental knowledge has the potential to advance both the rational metabolic engineering of acetogen cell factories and the accurate *in silico* reconstruction of their phenotypes (1, 2, 64, 65). Our study also highlights the need for large-scale mapping of genotype-phenotype relationships in acetogens to infer *in vivo* functionalities of isoenzymes and proteins with unknown or unclear functions. This absolute proteomics data set serves as a reference toward a better systems-level understanding of the ancient metabolism of acetogens.

## MATERIALS AND METHODS

**Bacterial strain and culture growth conditions.** Absolute proteome quantification was performed from high-biomass-concentration (~1.4 gDCW/L) steady-state autotrophic chemostat cultures of *C. autoethanogenum* growing on three different gas mixtures under culturing conditions described in our previous works (18, 19). Briefly, four biological replicate chemostat cultures of *C. autoethanogenum* strain DSM 19630 were grown on a chemically defined medium (without yeast extract) on either CO (~60% CO and 40% Ar), syngas (~50% CO, 20% $H_2$, 20% $CO_2$, and 10% $N_2$/Ar), or CO+$H_2$, termed "high-$H_2$ CO" (~15% CO, 45% $H_2$, and 40% Ar), under strictly anaerobic conditions. The bioreactors were maintained at 37°C, pH 5, and a dilution rate of ~1 day$^{-1}$ (specific growth rate of ~0.04 h$^{-1}$).

**Cell-free synthesis of stable-isotope-labeled protein standards.** Twenty proteins covering *C. autoethanogenum* central carbon metabolism, the HytA-E hydrogenase, and a ribosomal protein (see Table S1 in the supplemental material) were selected for the cell-free synthesis of SIL-proteins as described previously (18). Briefly, genes encoding these proteins were synthesized by commercial gene synthesis services (Biomatik). Target genes were subcloned into the cell-free expression vector pEUE01-His-N2 (CellFree Sciences) and transformed into *Escherichia coli* DH5$\alpha$ from which plasmid DNA was extracted and purified. The correct gene insertion into pEUE01-His-N2 was verified by DNA sequencing. Subsequently, the cell-free synthesis of His tag-fused *C. autoethanogenum* proteins was performed using the bilayer reaction method with the wheat germ extract WEPRO8240H (CellFree Sciences) as described previously (44, 45). mRNAs for cell-free synthesis were prepared by an *in vitro* transcription reaction, while the *in vitro* translation of target proteins was performed using a bilayer reaction where the translation layer was supplemented with L-Arg-$^{13}C_6$,$^{15}N_4$ and L-Lys-$^{13}C_6$,$^{15}N_2$ (Wako) at final concentrations of 20 mM to achieve high efficiency (>99%) for stable-isotope labeling of proteins. The *in vitro*-synthesized SIL-protein sequences also contained an N-terminal amino acid sequence, GYSFTTTAEK, that was later used as a tag for the quantification of the SIL-protein stock concentration. Subsequently, SIL-proteins were purified using Ni-Sepharose high-performance resin (GE Healthcare Life Sciences) and precipitated using methanol-chloroform-water precipitation in Eppendorf protein LoBind tubes. Finally, precipitated SIL-proteins were reconstituted in 104 $\mu$L of 8 M urea (UA; Sigma-Aldrich) in 0.1 M Trizma base (pH 8.5) by vigorous vortexing and stored at −80°C until further use.

**Absolute quantification of SIL-protein standards using PRM MS.** Concentrations of the 20 synthesized SIL-protein standard stocks were determined using PRM MS preceded by in-solution digestion of proteins and sample desalting and preparation for MS analysis.

**(i) Sample preparation.** Only Eppendorf protein LoBind tubes and pipette tips were used for all sample preparation steps. First, 20 $\mu$L of UA was added to 4 $\mu$L of the SIL-protein standard stock used to

determine the stock concentration, and the mix was vortexed. Next, 1 $\mu$L of 0.2 M dithiothreitol (DTT; Promega) was added, followed by vortexing and incubation for 1 h at 37°C to reduce disulfide bonds. Sulfhydryl groups were alkylated with 2 $\mu$L of 0.5 M iodoacetamide (IAA; Sigma-Aldrich), vigorously vortexed, and incubated for 30 min at room temperature in the dark. Next, 75 $\mu$L of 25 mM ammonium bicarbonate was added to dilute UA down to a 2 M concentration. Subsequently, 2 pmol (2 $\mu$L of stock) of the nonlabeled Aqua peptide HLEAAKGYSFTTTAEKAAELHK (Sigma-Aldrich) containing the quantification tag sequence GYSFTTTAEK was added to enable the quantification of SIL-protein stock concentrations using MS analysis based on the ratio of heavy to light GYSFTTTAEK signals (see below). Protein digestion was performed for 16 h at 37°C with 0.1 $\mu$g of trypsin–Lys-C mix (1 $\mu$L of stock; Promega) and stopped by lowering the pH to 3 by adding 5 $\mu$L of 10% (vol/vol) trifluoroacetic acid (TFA).

Samples were desalted using $C_{18}$ ZipTips (Merck Millipore) as follows: the column was wetted using 0.1% (vol/vol) formic acid (FA) in 100% acetonitrile (ACN), equilibrated with 0.1% FA in 70% (vol/vol) ACN, and washed with 0.1% FA before loading the sample and washing again with 0.1% FA. Peptides were eluted from the ZipTips with 0.1% FA in 70% ACN. Finally, samples were dried using a vacuum centrifuge (Eppendorf) at 30°C until dryness, followed by reconstitution in 12 $\mu$L of 0.1% FA in 5% ACN for subsequent MS analysis.

**(ii) LC method for PRM MS.** A Thermo Fisher Scientific UltiMate 3000 RSLCnano ultrahigh-performance liquid chromatography (UHPLC) system was used to elute the samples. Each sample was initially injected (6 $\mu$L) onto a Thermo Fisher Acclaim PepMap $C_{18}$ trap reversed-phase column (300-$\mu$m by 5-mm nano viper, 5-$\mu$m particle size) at a flow rate of 15 $\mu$L/min using 2% ACN for 3 min, with the solvent going to waste. The trap column was switched inline with the separation column (Grace Vydac Everest $C_{18}$, 300 Å, 150 $\mu$m by 150 mm, 2 $\mu$m), and the peptides were eluted at a flow rate of 3 $\mu$L/min using 0.1% FA in water (buffer A) and 80% ACN in buffer A (buffer B) as the mobile phases for gradient elution. Following 3 min, isocratic, of 3% buffer B, peptide elution employed a 3 to 40% ACN gradient for 28 min, followed by 40 to 95% ACN for 1.5 min and 95% ACN for 1.5 min at 40°C. The total elution time was 50 min, including a 95% ACN wash and a reequilibration step.

**(iii) PRM MS data acquisition.** The eluted peptides from the $C_{18}$ column were introduced to the MS system via a nano-electrospray ionization (ESI) source and analyzed using the Thermo Fisher Scientific Q-Exactive HF-X mass spectrometer. The electrospray voltage was 1.8 kV in positive-ion mode, and the ion transfer tube temperature was 250°C. Full MS scans were acquired in the Orbitrap mass analyzer over the range of $m/z$ 550 to 560 with a mass resolution of 30,000 (at $m/z$ 200). The automatic gain control (AGC) target value was set at 1.00E+06, and the maximum accumulation time was 50 ms for full MS scans. The PRM inclusion list included two mass values of 552.7640 and 556.7711. Tandem MS (MS/MS) spectra were acquired in the Orbitrap mass analyzer with a mass resolution of 15,000 (at $m/z$ 200). The AGC target value was set at 1.00E+06, and the maximum accumulation time was 30 ms for MS/MS, with an isolation window of 2 $m/z$. The loop count was set at 14 to gain greater MS/MS data.

**(iv) PRM MS data analysis.** Analysis of PRM MS data was performed using Skyline software (48). The following parameters were used to extract PRM MS data for the quantification tag sequence GYSFTTTAEK: three precursor isotope peaks with a charge of 2 (++) were included (monoisotopic; M + 1; M + 2), five of the most intense y product ions from ion 3 to the last ion of charge states 1 and 2 among the precursor were picked, chromatograms were extracted with an ion match mass tolerance of 0.05 $m/z$ for product ions by including all matching scans, full trypsin specificity was used with two missed cleavages allowed for peptides with a length of 8 to 25 amino acids (aa), and cysteine carbamidomethylation was used as a fixed peptide modification. Additionally, peptide modifications included heavy labels for lysine and arginine as $^{13}C_6{}^{15}N_2/+8.014$ Da (K) and $^{13}C_6{}^{15}N_4/+10.008$ Da (R), respectively. This translated into the SIL-proteins and the nonlabeled Aqua peptide possessing the tag GYSFTTTAEK with $m/z$ 556.7711 and 552.7640, respectively. Hence, the concentrations of SIL-protein stocks were calculated based on the ratio of heavy to light GYSFTTTAEK signals and the spike-in of 2 pmol of the nonlabeled Aqua peptide (see above). The high accuracy of quantification was evidenced by the very high similarity between both precursor peak areas and the expected isotope distribution ($R^2 > 0.99$ [idotp in Skyline]) and the heavy and light peak areas ($R^2 > 0.99$ [rdotp in Skyline]) for all SIL-protein standard stocks. No heavy GYSFTTTAEK signal was detected for protein CAETHG_RS16140 (an acetylating acetaldehyde dehydrogenase in the NCBI annotation of the sequence under GenBank accession number NC_022592.1 [69]); thus, 19 SIL-proteins could be used for the following absolute proteome quantification in *C. autoethanogenum* (Table S1).

**Absolute proteome quantification in *C. autoethanogenum* using DIA MS.** We used 19 synthetic heavy SIL variants (see above) of key *C. autoethanogenum* proteins (Table S1) as spike-in standards for the quantification of intracellular concentrations of their non-SIL counterparts using a DIA MS approach (46). Also, we performed a dilution series experiment for the spike-in SIL-proteins to ensure accurate absolute quantification. We refer to these 19 intracellular proteins as anchor proteins that were further used to estimate proteome-wide absolute protein concentrations in *C. autoethanogenum*. This was achieved by determining the best linear fit between anchor protein concentrations and their measured DIA MS intensities using the same strategy as the one described previously (28).

**(i) Preparation of spike-in SIL-protein standard mix and dilution series samples.** Only Eppendorf protein LoBind tubes and pipette tips were used for all preparation steps. The 19 spike-in SIL-protein standards that could be used for absolute proteome quantification in *C. autoethanogenum* (see above) were mixed in two lots: (i) "sample spike-in standard mix," with SIL-protein quantities matching estimated intracellular anchor protein quantities (i.e., expected light-to-heavy [L/H] ratios of ~1) based on label-free absolute quantification of the same samples in our previous work (18), and (ii) "dilution series

standard mix," with SIL-protein quantities doubling the estimated intracellular anchor protein quantities for the dilution series sample with the highest SIL-protein concentrations.

To ensure accurate absolute quantification of anchor protein concentrations, a dilution series experiment was performed to determine the linear dynamic quantification range and LLOQ for each of the 19 spike-in SIL-proteins. Dilution series samples were prepared by making nine 2-fold dilutions of the dilution series spike-in standard mix (i.e., 10 samples total for dilution series, with a 512-fold concentration span) in a constant $C.$ $autoethanogenum$ cell lysate background (0.07 $\mu$g/$\mu$L; 10 $\mu$g/tube) serving as a blocking agent to avoid the loss of purified SIL-proteins (to container and pipette tip walls) and as a background proteome for accurate MS quantification of the linear range and LLOQ for anchor proteins.

**(ii) Sample preparation.** $C.$ $autoethanogenum$ cultures were sampled for proteomics by immediately pelleting 2 mL of the culture by centrifugation (25,000 $\times$ $g$ for 1 min at 4°C) and stored at $-80$°C until analysis. Details of protein extraction and protein quantification in cell lysates were described previously (18). In short, thawed cell pellets were suspended in lysis buffer (containing SDS, DTT, and Trizma base), and cell lysis was performed by using glass beads and repeating a "lysis cycle" consisting of heating, bead beating, centrifugation, and vortexing before protein quantification using the 2D Quant kit (GE Healthcare Life Sciences).

Sample preparation and protein digestion for MS analysis were based on the filter-aided sample preparation (FASP) protocol (70). The following starting material was loaded onto an Amicon Ultra 0.5-mL centrifugal filter unit (nominal molecular weight cutoff of 30,000; Merck Millipore): (i) 50 $\mu$g of protein for one culture sample from each gas mixture (CO, syngas, or high-H$_2$ CO) for building the spectral library for DIA MS data analysis (samples 1 to 3), (ii) 7 $\mu$g of protein for one culture sample from either syngas or high-H$_2$ CO plus sample spike-in standard mix for including spike-in SIL-protein data in the spectral library (samples 4 and 5), (iii) 15 $\mu$g of protein for all 12 culture samples (biological quadruplicates from CO, syngas, and high-H$_2$ CO) plus sample spike-in standard mix for performing absolute proteome quantification in $C.$ $autoethanogenum$ (samples 6 to 17), and (iv) 10 dilution series samples with 10 to 15 $\mu$g of total protein ($C.$ $autoethanogenum$ cell lysate background plus dilution series spike-in standard mix) for performing the dilution series experiment for the 19 spike-in SIL-proteins (see above) (samples 18 to 27).

Samples containing SIL-proteins (samples 4 to 27) were incubated at 37°C for 1 h to reduce SIL-protein disulfide bonds (the cell lysate contained DTT). Details of the FASP workflow were described previously (18). In short, samples were washed with UA, sulfhydryl groups were alkylated with IAA, proteins were digested using a trypsin–Lys-C mix, and peptides were eluted from the filter with 60 $\mu$L of ammonium bicarbonate. Next, 50 $\mu$L of samples 1 to 3 was withdrawn and pooled for performing high-pH reverse-phase fractionation as described previously (18) for expanding the spectral library for DIA MS data analysis, yielding eight fractions (samples 28 to 35). Subsequently, all samples were vacuum centrifuged at 30°C until dryness, followed by reconstitution of samples 1 to 3 and 4 to 35 in 51 and 13 $\mu$L of 0.1% FA in 5% ACN, respectively. Finally, the total peptide concentration in each sample was determined using the Pierce quantitative fluorometric peptide assay (Thermo Fisher Scientific) to ensure that the same total peptide amount across samples 1 to 17 and 28 to 35 (excluding samples 18 to 27 [see below]) could be injected for DIA MS analysis.

**(iii) LC method for DDA and DIA MS.** Details of the liquid chromatography (LC) method employed for generating the spectral library using data-dependent acquisition (DDA) and for DIA sample runs were described previously (20). In short, a Thermo Fisher Scientific UHPLC system, including C$_{18}$ trap and separation columns, was used to elute peptides with a gradient and a total elution time of 110 min. For each DDA and DIA sample run, 1 $\mu$g of peptide material from protein digestion was injected, except for dilution series samples (samples 18 to 27 described above), which were injected in a constant volume of 3 $\mu$L to maintain the dilution levels of the dilution series spike-in standard mix.

**(iv) DDA MS spectral library generation.** The following 13 samples were analyzed on the Q-Exactive HF-X instrument in DDA mode to yield the spectral library for DIA MS data analysis: (i) three replicates of one culture sample from each gas mixture (CO, syngas, or high-H$_2$ CO) (samples 1 to 3 described above), (ii) three replicates of one culture sample from either syngas or high-H$_2$ CO plus sample spike-in standard mix (samples 4 and 5), and (iii) eight high-pH reverse-phase fractions of a pool of samples from each gas mixture (samples 28 to 35).

Details of DDA MS acquisition for generating the spectral library were described previously (20). In short, eluted peptides from the C$_{18}$ column were introduced to the MS system via a nano-ESI device and analyzed using the Q-Exactive HF-X mass spectrometer with an Orbitrap mass analyzer. The DDA MS spectral library for DIA MS data confirmation and quantification using Skyline software (48) was created using Proteome Discoverer 2.2 software (Thermo Fisher Scientific) and its SEQUEST HT search as described previously (18). The final .pd result file contained peptide-spectrum matches (PSMs) with $q$ values estimated at a 1% false discovery rate (FDR) for peptides of $\geq$4 aa.

**(v) DIA MS data acquisition.** Details of DIA MS acquisition were described previously (20). In short, as described above for DDA MS acquisition, eluted peptides were introduced to the MS system via a nano-ESI device and analyzed using the Q-Exactive HF-X mass spectrometer with an Orbitrap mass analyzer. DIA was achieved using an inclusion list of $m/z$ 395 to 1,100 in steps of 15 amu and scans cycled through the list of 48 isolation windows with a loop count of 48. In total, DIA MS data were acquired for 22 samples (samples 6 to 27 defined in the sample preparation section above).

**(vi) DIA MS data analysis.** DIA MS data analysis was performed with Skyline (48) as described previously (18), with the following modifications: (i) 12 manually picked high-confidence endogenous peptides present in all samples and spanning the elution gradient were used for internal retention time (iRT) alignment by building an RT predictor; (ii) outlier peptides from iRT regression were removed; (iii) a

minimum of three isotope peaks were required for a precursor; (iv) a single peptide per spike-in SIL-protein was allowed for anchor protein absolute quantification, while at least two peptides per protein were required for label-free estimation of proteome-wide protein concentrations; and (v) extracted ion chromatograms (XICs) were transformed using Savitzky-Golay smoothing. Briefly, the .pd result file from Proteome Discoverer was used to build the DIA MS spectral library, the mProphet peak-picking algorithm (47) within Skyline was used to separate true- from false-positive peak groups (per sample), and only peak groups with a $q$ value of <0.01 (representing a 1% FDR) were used for further quantification. We confidently quantitated 7,288 peptides and 1,243 proteins across all samples and 4,887 peptides and 1,043 proteins on average within each sample for estimating proteome-wide absolute protein concentrations. For the absolute quantification of anchor protein concentrations, we additionally manually removed (i) the integration of peaks showing non-Gaussian shapes or interference from other peaks and (ii) precursors with similarity measures of an $R^2$ of <0.9 between product peak areas and the corresponding intensities in the spectral library (dotp in Skyline), precursor peak areas and the expected isotope distribution (idotp), or heavy and light peak areas (rdotp). After analysis in Skyline, 17 spike-in SIL-proteins remained for further analysis as protein CAETHG_RS14410 was not identified in the DIA MS data, while CAETHG_RS18395 did not pass quantification filters (Table S1).

(vii) Absolute quantification of anchor protein concentrations. We employed further stringent criteria on top of the output from Skyline analysis to ensure high-confidence absolute quantification of 17 anchor protein concentrations. First, the precursor with the highest heavy intensity for the highest dilution series spike-in standard mix sample in the dilution series (DS01) was kept, while the others were deleted. Peptides quantified in fewer than three biological replicate cultures within a gas mixture, with no heavy signal for the DS01 sample, or with heavy signals for fewer than three continuous dilution series samples were removed. Next, we utilized the dilution series experiment to keep only signals over the LLOQ and within the linear dynamic quantification range. For this, a correlation between experimental and expected peptide L/H signal ratios for each peptide across the dilution series was made to determine the LLOQ and calculate the correlation, slope, and intercept between the MS signal and the spike-in level (Table S2). Only peptides showing a correlation at $R^2$ values of >0.95, 0.95 < slope < 1.05, and $-0.1 <$ intercept $< 0.1$ for the dilution series were kept. This ensured that we were using only peptides within the linear dynamic range. The remaining peptides were further filtered for each culture sample by removing peptides whose light or heavy signal was below the LLOQ in the dilution series. Subsequently, only peptides with L/H ratios for at least three biological replicate cultures for each gas mixture were kept (i.e., ≥9 data points). Finally, we aimed to detect outlier peptides by calculating the percent difference of a peptide's L/H ratio from the average L/H ratio of all peptides for a given protein for every sample. Peptides were considered outliers and thus removed if the average difference across all samples was >50% or if the average difference within biological replicate cultures was >50%. After the above-described stringent criteria were applied, 106 high-confidence peptides remained (Table S2) for the quantification of 16 anchor protein concentrations since CAETHG_RS01830 was lost during manual analysis (Table 1 and Table S1). Proteins CAETHG_RS13725 and CAETHG_RS07840 were excluded from the high-$H_2$ CO culture data set as their calculated concentrations varied >50% between biological replicates. Data for one high-$H_2$ CO culture were excluded from further analysis due to differences from bioreplicates, likely due to challenges with MS analysis.

(viii) Label-free estimation of proteome-wide protein concentrations. We used the anchor proteins to estimate proteome-wide protein concentrations in *C. autoethanogenum* by determining the best linear fit between anchor protein concentrations and their measured DIA MS intensities using the aLFQ R package (49) and the same strategy as the one described previously for SWATH MS (28). Briefly, aLFQ used anchor proteins and cross-validated model selection by bootstrapping to determine the optimal model within various label-free absolute proteome quantification approaches (e.g., TopN and iBAQ). The approach can obtain the model with the smallest error between anchor protein concentrations determined using SIL-protein standards and label-free estimated concentrations. The models with the highest accuracy were used to estimate proteome-wide label-free concentrations for all proteins from their DIA MS intensities (1,043 proteins on average within each sample, with minimally two peptides per protein [see above]), summing the five most intense fragment ion intensities of the most or three of the most intense peptides per protein for CO or high-$H_2$ CO cultures, respectively, and summing the five most intense fragment ion intensities of all quantified peptides of the protein divided by the number of theoretically observable peptides (i.e., iBAQ [71]) for syngas cultures.

Expected protein complex stoichiometries. Equimolar stoichiometries for the HytA-E/FdhA and MetFV protein complexes were expected based on SDS gel staining experiments in *C. autoethanogenum* (57) and the acetogen *Moorella thermoacetica* (72), respectively. Expected stoichiometries for other protein complexes in Fig. 3 were based on measured stoichiometries in *E. coli* K-12 (Complex Portal [www.ebi.ac.uk/complexportal]) and significant homology between complex member proteins in *C. autoethanogenum* and *E. coli*. All depicted *C. autoethanogenum* protein complex members had NCBI protein-protein BLAST E values of $<10^{-16}$ and scores of >73 against the respective *E. coli* K-12 proteins using non-redundant protein sequences.

Generation of proteomaps. The distribution of proteome-wide protein concentrations among functional gene classifications was visualized using proteomaps (27). For this, the NCBI annotation of the sequence under GenBank accession number NC_022592.1 (69) was used as the annotation genome for *C. autoethanogenum*, with CAETHG_RS07860 being removed and replaced with the carbon monoxide dehydrogenase genes named CAETHG_RS07861 and CAETHG_RS07862 with initial identifications of CAETHG_1620 and CAETHG_1621, respectively. Functional categories were assigned to protein sequences with KO IDs (51) using the KEGG annotation tool BlastKOALA (73). Since proteomaps require a tree-like hierarchy, proteins that were automatically assigned to multiple functional categories were manually

assigned to one bottom-level category (level 3 in Table S4) based on their principal task. We also created the functional categories "C$_1$ fixation/Wood-Ljungdahl pathway" (level 2/3), "acetate and ethanol synthesis" (level 3), "energy conservation" (level 3), and "hydrogenases" (level 3) and manually assigned key acetogen proteins to these categories to reflect more accurately the functional categories for an acetogen. Proteins without designated KO IDs were manually assigned to categories or grouped under "not included in pathway or Brite" (level 1) with levels 2 and 3 as "no KO ID." If BlastKOALA assigned multiple genes the same proposed gene/protein name, unique numbers were added to names (e.g., PfKA and PfkA2). The final "treemap" defining the hierarchy for our proteomaps is in Table S4.

**Intracellular metabolic flux rates estimated by genome-scale metabolic modeling.** In this work, we used previously estimated intracellular specific metabolic flux rates (mmol/gDCW/h) for the same *C. autoethanogenum* CO, syngas, and high-H$_2$ CO cultures as the ones studied here (18). Briefly, Valgepea et al. simulated flux data using the genome-scale metabolic model iCLAU786 of *C. autoethanogenum* (18) and flux balance analysis (FBA) (74). The model was constrained with experimentally measured data for gas uptake and production (CO, CO$_2$, and H$_2$), cysteine uptake, by-product secretion (acetate, ethanol, and 2,3-butanediol), and cellular specific growth rate. Maximization of ATP dissipation was used as the objective function to perform FBA calculations using the COBRA toolbox (75) as a programming platform with Gurobi (Gurobi Optimization Inc.) as the linear programming solver. See reference 18 for complete simulation details and all intracellular fluxes [simulations SIM10-19 in Table S4 in reference 18]. In this work, we performed flux variability analysis (FVA) using the COBRA toolbox with CPLEX (IBM) as the solver to validate the FBA-estimated fluxes used for flux regulation analysis and confirmed that the fluxes are within the FVA flux ranges (Table S6).

**Calculation of apparent *in vivo* catalytic rates of enzymes.** We calculated the apparent *in vivo* catalytic rates of enzymes, denoted $k_{app}$ (s$^{-1}$) (26), as the ratio of the specific flux rate (mmol/gDCW/h) to the protein concentration (nmol/gDCW). Gene-protein-reaction (GPR) data of the genome-scale metabolic model iCLAU786 (18) were manually curated to reflect the most recent knowledge and used to link metabolic fluxes with catalyzing enzymes. For reactions with multiple assigned enzymes (i.e., isoenzymes), the enzyme with the highest average ranking of its concentration across the three cultures (Table S3) was assumed to solely catalyze the flux. For enzyme complexes, the average of quantified subunit concentrations was used (standard deviation estimated using error propagation). For the HytA-E hydrogenase, its measured protein concentration was split between reactions "rxn08518_c0" (direct CO$_2$ reduction with H$_2$ in complex with FdhA) and "leq000001_c0" (H$_2$ oxidation solely by HytA-E) proportionally to flux for syngas and high-H$_2$ CO cultures. The resulting enzymes or enzyme complexes catalyzing specific fluxes are shown in Table S5. Finally, we assumed each protein chain to be catalytically active and calculated $k_{app}$ values only for metabolic reactions with a nonzero flux under at least one condition, a specific flux rate of >0.1% of CO fixation flux under at least one condition, and label-free data with a measured concentration for the associated enzyme(s) under all conditions (Table S5). Membrane proteins were excluded from $k_{app}$ calculations to avoid bias from potentially incomplete protein extraction. This produced $k_{app}$ values for 48 enzymes/complexes using label-free protein concentrations and for 13 of those also using anchor protein concentrations (Table S5).

**Determination of regulation levels of metabolic fluxes.** We used relative proteomics data from the same work as that for the flux data (18) that studied the same cultures as the ones in this work to determine whether fluxes are regulated by changing enzyme concentrations or their catalytic rates. We considered metabolic fluxes with nonzero specific flux rates under at least two conditions of CO, syngas, or high-H$_2$ CO cultures. The same manually curated GPRs and criteria for isoenzymes and protein complexes as the ones described above for $k_{app}$ calculation were used to determine flux-enzyme pairs (Table S6). We first used two-tailed two-sample equal-variance Student's $t$ test with FDR correction (76) to determine fluxes with significant changes between any two conditions ($q$ value of <0.05). We then used Student's left-tailed $t$ distribution with FDR to determine if the significant flux change for every flux was significantly different from the change in the respective enzyme expression between the same conditions (Table S6). Flux with a $q$ value of <0.05 for the latter test was considered to be regulated at the posttranslational level (e.g., by changing enzyme catalytic rate).

**Data availability.** All data generated or analyzed during this study are in the text, supplemental material, or public databases. Raw PRM MS data have been deposited to Panorama at https://panoramaweb.org/Valgepea_Cauto_PRM.url with ProteomeXchange Consortium data set identifier PXD025760. Raw DIA MS data have been deposited to the ProteomeXchange Consortium via the PRIDE partner repository (77) with the data set identifier PXD025732. PRM MS data with all Skyline processing settings can be viewed and downloaded from Panorama at https://panoramaweb.org/Valgepea_Cauto_PRM.url. DIA MS data with all Skyline processing settings can be viewed and downloaded from Panorama at https://panoramaweb.org/Valgepea_Cauto_Anchors.url for anchor protein absolute quantification and at https://panoramaweb.org/Valgepea_Cauto_LF.url for estimating proteome-wide absolute protein concentrations.

## SUPPLEMENTAL MATERIAL

Supplemental material is available online only.

**FIG S1**, PDF file, 0.9 MB.

**FIG S2**, PDF file, 0.1 MB.

**FIG S3**, PDF file, 0.2 MB.

**FIG S4**, PDF file, 0.5 MB.

**TABLE S1**, XLSX file, 0.01 MB.

**TABLE S2**, XLSX file, 0.1 MB.
**TABLE S3**, XLSX file, 0.2 MB.
**TABLE S4**, XLSX file, 0.2 MB.
**TABLE S5**, XLSX file, 0.05 MB.
**TABLE S6**, XLSX file, 0.04 MB.

## ACKNOWLEDGMENTS

We thank Jörg Bernhardt for help with proteomaps, Tim McCubbin for scripts, Andrus Seiman for statistics, and Olivier Lemaire for valuable discussions.

This work was funded by the Australian Research Council (ARC LP140100213) in collaboration with LanzaTech. We thank the following investors in LanzaTech's technology: Stephen Tindall, Khosla Ventures, Qiming Venture Partners, Softbank China, the Malaysian Life Sciences Capital Fund, Mitsui, Primetals, CICC Growth Capital Fund I, L.P., and the New Zealand Superannuation Fund. The research utilized equipment and support provided by the Queensland node of Metabolomics Australia, an initiative of the Australian Government being conducted as part of the NCRIS National Research Infrastructure for Australia. There was no funding support from the European Union and the Estonian Research Council for the experimental part of the study. However, Kaspar Valgepea also acknowledges support from the European Union's Horizon 2020 Research and Innovation Programme under grant agreement N810755 and the Estonian Research Council's grant agreement PSG289.

Kaspar Valgepea, Conceptualization, Methodology, Formal Analysis, Investigation, Writing–Original Draft, and Writing–Review & Editing; Gert Talbo, Methodology, Formal Analysis, and Investigation; Nobuaki Takemori, Methodology, Investigation, Resources, and Writing–Review & Editing; Ayako Takemori, Methodology, Investigation, and Resources; Christina Ludwig, Conceptualization, Methodology, and Writing–Review & Editing; Vishnuvardhan Mahamkali, Formal Analysis and Writing–Review & Editing; Alexander P. Mueller, Methodology and Writing–Review & Editing; Ryan Tappel, Methodology, Writing–Review & Editing, and Project Administration; Michael Köpke, Conceptualization, Resources, Writing–Review & Editing, Supervision, and Funding Acquisition; Séan Dennis Simpson, Resources and Funding Acquisition; Lars Keld Nielsen, Conceptualization, Resources, Writing–Review & Editing, Supervision, and Funding Acquisition; Esteban Marcellin, Conceptualization, Methodology, Resources, Writing–Original Draft, Writing–Review & Editing, Supervision, Project Administration, and Funding Acquisition.

LanzaTech has interest in commercial gas fermentation with *C. autoethanogenum*. Vishnuvardhan Mahamkali, Alexander P. Mueller, Ryan Tappel, Michael Köpke, and Séan Dennis Simpson are employees of LanzaTech.

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
