## [Reviewer comments · mSystems]

Absolute proteome quantification in the gas-fermenting acetogen *Clostridium autoethanogenum*

Kaspar Valgepea, Gert Talbo, Nobuaki Takemori, Ayako Takemori, Christina Ludwig, Vishnuvardhan Mahamkali, Alexander Mueller, Ryan Tappel, Michael Köpke, Sean Simpson, Lars Nielsen, and Esteban Marcellin

Corresponding Author(s): Esteban Marcellin, University of Queensland

Review Timeline:

Submission Date:	January 17, 2022
Editorial Decision:	February 22, 2022
Revision Received:	March 8, 2022
Accepted:	March 10, 2022

Editor: Karoline Faust

Reviewer(s): Disclosure of reviewer identity is with reference to reviewer comments included in decision letter(s). The following individuals involved in review of your submission have agreed to reveal their identity: Diana Zita Sousa (Reviewer #1); Byung-Kwan Cho (Reviewer #4)

Transaction Report:

DOI: <https://doi.org/10.1128/msystems.00026-22>

February 22, 2022

Dr. Esteban Marcellin
University of Queensland
Australian Institute for Bioengineering and Nanotechnology (AIBN)
St Lucia, QLD 4072, Australia
Brisbane, Brisbane QLD 4072
Australia

Re: mSystems00026-22 (Absolute proteome quantification in the gas-fermenting acetogen *Clostridium autoethanogenum*)

Dear Dr. Esteban Marcellin:

Thank you for submitting your manuscript to mSystems. I am pleased to inform you that, in principle, we expect to accept it for publication in mSystems. However, acceptance will not be final until you have adequately addressed the reviewer comments.

Below you will find instructions from the mSystems editorial office and comments generated during the review.

Preparing Revision Guidelines

Sincerely,

Karoline Faust

Editor, mSystems

Journals Department
Reviewer comments:

Reviewer #1 (Comments for the Author):

The authors addressed adequately the comments raised in previous revision. Description of the methodology and data analysis has been substantially improved in this version of the manuscript.

Reviewer #3 (Comments for the Author):

In this paper, the authors performed absolute proteome quantification under three autotrophic conditions (CO, syngas, or high H₂+CO) of *C. autoethanogenum*, and revealed the main metabolic pathways and enzymes. Besides, they provided valuable information of enzyme concentrations, kinetics of major metabolic pathways under three autotrophic condition comparing their proteome information with the flux obtained from the genome-scale model. These results will be very helpful in understanding the acetogen in autotrophic conditions at the system-level, and it will also be very helpful in the study of the genome-scale metabolic model.

My comments can be found as below;

- 1) Line 100, Please explain the reason why the proteome of the *C. autoethanogenum* strain was performed under three autotrophic conditions. Also, the phenotypes (e.g. growth rate, acetate, or ethanol production) of *C. autoethanogenum* under three autotrophic conditions must be described and compared to understand the genotype-phenotype relationship claimed by the authors.
- 2) Line 104, Table S1, Please explain what is the criteria for selecting the 19 synthetic heavy SIL-variant in Table S1.
- 3) Line 193, section 4, ~Line 193, section 5, It is difficult to understand whether the main enzyme concentration of WLP is an autotrophic-specific proteome because only the results under autotrophic condition are shown.
- 4) Line 193, Figure 6, The authors have mentioned the importance of CODH in the section 5. So, in Figure 6, the major genes of WLP, especially the change of CODH ratio in 3 conditions, seems to help the reader in understanding.
- 5) Line 193, Section 6, It is very unfortunate that the energy conservation part is missing. There is no information on membrane proteins such as Rnf complex or ATP synthase. Was the proteome information of membrane proteins fraction unknown?
- 6) Line 228, Section 7, The authors described the strength of each enzyme reaction with enzyme concentration, GEMs flux, and Kapps value. However, like Rubisco in plants, there may be cases where the enzyme concentration is high but the Kapps value is low. Please describe any such results in the authors' data.
- 7) Line 197, There are cases where several isoenzymes (e.g., hydrogenase, alcohol dehydrogenase) are involved in one metabolic reaction. Please describe how you calculated Kapps.
- 8) Line 219~222, the ratio of AcsCD-toAcsB increases with the CO-Syngas-high H₂CO condition. However, the concentration of CO gas is the lowest under High H₂CO conditions. If the flux increases as the CODH ratio increases, please explain whether there is a correlation with the actual measured value of the CO oxidation rate.

March 10, 2022

Dr. Esteban Marcellin
University of Queensland
Australian Institute for Bioengineering and Nanotechnology (AIBN)
St Lucia, QLD 4072, Australia
Brisbane, Brisbane QLD 4072
Australia

Re: mSystems00026-22R1 (Absolute proteome quantification in the gas-fermenting acetogen *Clostridium autoethanogenum*)

Dear Dr. Esteban Marcellin:

It is my pleasure to inform you that your manuscript has been accepted and that I am forwarding it to the ASM Journals Department for publication. For your reference, ASM Journals' address is given below. Before it can be scheduled for publication, your manuscript will be checked by the mSystems production staff to make sure that all elements meet the technical requirements for publication. They will contact you if anything needs to be revised before copyediting and production can begin. Otherwise, you will be notified when your proofs are ready to be viewed.

Publication Fees:

We recognize that the video files can become quite large, and so to avoid quality loss ASM suggests sending the video file via <https://www.wetransfer.com/>. When you have a final version of the video and the still ready to share, please send it to mSystems staff at mssystemsjournal@msubmit.net.

For mSystems research articles, if you would like to submit an image for consideration as the Featured Image for an issue, please contact mSystems staff at mssystemsjournal@msubmit.net.

Sincerely,

Karoline Faust

Editor, mSystems

Journals Department
Table S4: Accept

Fig. S2: Accept

Table S3: Accept

Table S2: Accept

Fig. S3: Accept

Table S6: Accept

Table S5: Accept

Table S1: Accept

Fig. S1: Accept

Fig. S4: Accept